

# Polyphenol-rich *Trapa quadrispinosa* pericarp extract ameliorates high-fat diet induced non-alcoholic fatty liver disease by regulating lipid metabolism and insulin resistance in mice

Tunyu Jian[1], Han Lü[1], Xiaoqin Ding[1], Yuexian Wu[1], Yuanyuan Zuo[1], Jiawei Li[1], Jian Chen[1,2] and Hong Gu[3]

[1] Institute of Botany, Jiangsu Province and Chinese Academy of Sciences, Nanjing, China
[2] Department of Food Science and Technology, College of Light Industry and Food Engineering, Nanjing Forestry University, Nanjing, China
[3] Department of Colorectal Surgery, Jiangyin Hospital of Traditional Chinese Medicine, Jiangyin, China

Corresponding authors
Han Lü, xiaohan1814@163.com
Jian Chen, chenjian80@aliyun.com

## ABSTRACT

In China, *Trapa quadrispinosa* (also called water caltrop) has long been used as a function food and folk medicine to treat diabetes mellitus for years. In the present study, the extract of *T. quadrispinosa* pericarp (TQPE) which mainly contains hydrolysable tannins was prepared to investigate the potential therapeutic action in non-alcoholic fatty liver disease (NAFLD) mice induced by high fat-diet (HFD). After the administration of TQPE (15, 30 mg/kg/day) for 8 weeks, the increased weight of body and liver were significantly suppressed. TQPE also ameliorated liver lipid deposition and reduced lipids parameters of blood in mice. Moreover, TQPE attenuated oxidative stress and showed a hepatoprotective effect in mice. TQPE was also found to decrease the value of homeostatic model assessment for insulin resistance. In addition, TQPE administration increased the phosphorylation of AMP-activated protein kinase (AMPK) and Acetyl-CoA carboxylase (ACC) and inhibited sterol regulatory element-binding protein (SREBP) in the liver tissue. Meanwhile, TQPE elevated insulin receptor substrate-1 (IRs-1) and protein kinase B (Akt) phosphorylation. These results reflected that, as a nature product, TQPE is a potential agent for suppressing the process of NAFLD via regulation of the AMPK/SREBP/ACC and IRs-1/Akt pathways.

## INTRODUCTION

As a common chronic liver disease, non-alcoholic fatty liver disease (NAFLD) is defined by pathological accumulation of lipid in the liver without excess alcohol consumption (*Golabi, Bush & Younossi, 2017*). Being a hepatic manifestation of metabolic syndrome, it is similar to those chronic metabolic disorders, such as obesity, insulin resistance, type 2 diabetes mellitus (T2DM), inflammation and cardiovascular disease (*Bagherniya et al., 2018*). NAFLD increases the risk of progressive liver injury, which appears as a continuum

disease progression, from simple steatosis to liver failure and hepatocellular carcinoma (*Suolang et al., 2019*). NAFLD has emerged as a worldwide serious public health burden, epidemiology of NAFLD have highlighted surprisingly high prevalence in many countries (the estimated prevalence is 25–30% in adults) (*Moore, 2019*; *Ratziu, 2018*). Therefore, there is a great demand for exploring effective therapeutic agents to treat and prevent NAFLD.

The recent evidence indicated that fat accumulation and insulin resistance (IR) are intensely associated with the development and progression of NAFLD (*Araujo et al., 2018*; *Fan et al., 2018*; *Jian et al., 2018*). As a highly evolutionarily conserved sensor of cellular energy status, AMP-activated protein kinase (AMPK) plays a critical role in regulating hepatic lipid metabolism including lipolysis, glucose transport and gluconeogenesis (*Brown & Goldstein, 1997*). Sterol regulatory element-binding protein (SREBP), a key transcription factor in regulating liver lipid synthesis, is the downstream of AMPK (*Li et al., 2011*). Acetyl-CoA carboxylase (ACC), a member of lipogenic factor, is the downstream target of SREBP. AMPK activation phosphorylates and inhibits ACC in adipose and hepatic tissues thus downregulate fatty acid synthesis (*Bijland, Mancini & Salt, 2013*; *Zhang, Xie & Leung, 2018*). In the NAFLD models of many studies, it was observed that the inhibition of phosphorylation of AMPK led to lipid accumulation by increasing SREBP and inhibiting ACC phosphorylation (*Chen et al., 2019*; *Li et al., 2018b*; *Park et al., 2019*; *Zhou et al., 2017*). In addition, IR is also strongly associated with hepatic lipid accumulation in NAFLD. Insulin signaling transduction is dependent on insulin receptor substrate-1 (IRs-1), and phosphorylation of IRs-1 give rise to insulin pathway activation (*Fu, Cui & Zhang, 2018*; *Saez-Lara et al., 2016*). Moreover, for insulin signaling cascade conduction, Protein kinase B (Akt) is another essential factor. Impairment of Akt activity has been demonstrated under NAFLD condition, thus activated Akt (increased phosphorylation) could ameliorate hepatic steatosis and improve IR in NAFLD model (*Fan et al., 2018*; *Jung et al., 2018*). Therefore, targeting regulation of AMPK and insulin signaling pathway might be a new and useful therapeutic approach to drop lipid accumulation and insulin resistance in NAFLD.

Nowadays, pharmacological studies have significantly expanded to screen natural products for exploration of novel pharmaceutical agents. Many studies revealed that medicinal plant extracts, herb formulas have remarkable therapeutic effect on NAFLD (*Bagheriya et al., 2018*; *Chen et al., 2017*; *Li et al., 2018a*; *Suolang et al., 2019*). *Trapa quadrispinosa*, also called water chestnut or water caltrop, is a floating-leaf aquatic plant, which is commonly cultivated in the south of China, as well as in India and southeast Asia. The fruit of water caltrop, is a function food and folk medicine, which could be used to treat metabolic syndrome such as diabetes mellitus (DM). But the pericarps of *T. quadrispinosa* were usually discarded in large quantities after the seeds had been harvested. Interestingly, recent researches have demonstrated that the pericarps of water caltrop also displayed multiple biological activities, including hypoglycemic (*Huang et al., 2016*), anti-tumor (*Lin et al., 2013*), anti-inflammatory (*Kim et al., 2015*), anti-oxidant effects and hepatprotective activity (*Kim et al., 2014*). To our knowledge, the therapeutic effect of *Trapa quadrispinosa* pericarps extract (TQPE) in high-fat diet (HFD) induced NAFLD, remains unknown.
The purpose of the present study was designed to determine whether *Trapa quadrispinosa* pericarps extract (TQPE) could attenuate NAFLD induced by HFD in mice, and also to explore a possible mechanism of this action.

## MATERIALS & METHODS

### Chemicals and reagents

HPLC grade methanol used for the mobile phase in HPLC-DAD/QTOF analysis was obtained from Tedia Co. Inc. (Fairfield, OH, USA). Formic acid in HPLC grade was taken from Acros Organics (Geel, Belgium). Pure water was produced from a Milli-Q system (Millipore, Bedford, MA, USA). Five standard compounds isolated from *T. quadrispinosa* were prepared and identified according to our previous study (*Lv, Jian & Ding, 2019*). Gallic acid was obtained from National Institutes for Food and Drug Control of China (Beijing, China). The other reagents were obtained from Sinopharm Chemical Reagent Co. Ltd. (Shanghai, China). A kit from KeyGen Biotechnology (Nanjing, China) was used for the protein extraction and BCA protein assay. Besides, the kits for detecting blood glucose, high-density lipoprotein cholesterol (HDL-c), low-density lipoprotein cholesterol (LDL-c), total cholesterol (TC), triglycerides (TG), aspartate aminotransferase (AST), alanine aminotransferase (ALT), superoxide dismutase (SOD) and malonaldehyde (MDA) were all obtained from Nanjing Jiancheng Bioengineering Institute (Nanjing, China). Determination of insulin by enzyme-linked immunosorbent assay (ELISA) commercially kits were purchased from Beyotime Institute of Biotechnology (Haimen, China).

The antibodies of AMPK$\alpha$, p-AMPK$\alpha$ (Thr172), ACC, p-ACC, IRs-, p-IRs-1 (Tyr895), Akt, p-Akt (Ser473) for western blotting (WB) were purchased from Cell Signaling Technology (Danvers, MA, USA). SREBP was obtained from Santa Cruz Biotechnology (Santa Cruz, USA). Anti-rabbit/anti-mouse IgG and HRP-linked antibody were purchased from Cell Signaling Technology (Danvers, USA).

### Preparation and characterization of *Trapa quadrispinosa* pericarp extract (TQPE)

The air-dried *T. quadrispinosa* (collected in Shandong Zaozhuang of China) pericarp powder (10.0 kg) was extracted twice with fifty litres 80% (v/v) ethanol by soaking at room temperature for 14 days. After concentrating in vacuum at 50 °C, the extract was suspended in distilled water and then partitioned by petroleum ether, ethyl acetate and normal-butanol successively. The ethyl acetate extract was first concentrated and then dissolved in water, and later applied to a column packed with macroreticular resin XAD16 (The Dow Chemical Company). The column was orderly eluted with a gradient of $H_2O$, 10%, 40%, 60% and 95% ethanol solution. The 40% ethanol elution was collected, concentrated and dried and finally the TQPE (175g) was obtained.

According to previous reported method, the total phenolic content of TQPE was measured by the Folin-Ciocalteu phenol reagent (*Singleton & Rossi, 1965*). Briefly, 1.0 mg of TQPE extract was solved in 1.0 ml of methanol was used as test solution. After incubation with Folin-Ciocalteu phenol reagent, the test solution was added 20% sodium carbonate to

develop a color and its absorbance was measured at 755 nm. The final value was expressed as gallic acid (standard) equivalent.

5.0 mg/ml of TQPE (dissolved in methanol) was injected in an Agilent 6530 accurate-mass quadrupole time-of-flight system (Agilent Technologies, CA, USA) for the HPLC-QTOF/MS analysis. Separation was conducted by an Agilent ZORBAX SB-C18 column (1.8 $\mu$m, 4.6 × 100 mm; Waldbronn, Germany). The mobile phase was composed of Methanol (A) and 0.1% formic acid (B) under gradient conditions (0–50 min, 10–50% A; 50–65 min, 50–100% A) at 1 ml/min. The QTOF-MS equipped with an electrospray ion source operated in the negative ion mode at 4 kV capillary voltage, 10 L/min drying gas and 350 °C gas temperature. Mass range of scanning was 100–1,000 m/z. MassHunter Qualitative Analysis software (Agilent Technologies, USA) was used to perform measurement and analysis.

Furthermore, HPLC-DAD was performed by Dionex Ultimate 3000 HPLC system (Thermo Fisher Scientific, USA) equipped with a diode array-UV detector at 210 nm. Nucleosil C18 column (4.6 mm × 150 mm, 3 $\mu$m, ThermoFisher Scientific, USA) was used for separation. The mobile phase was composed of Acetonitrile (A) and 0.05% trifluoroacetic acid (B) under gradient conditions as followed: 0–5 min, 5%A; 5–15 min, 5–10% A; 15–55 min, 10–25% A; 55–65 min, 25–100% A, the flow rate of HPLC was 1 ml/min.

## Animals
The male ICR mice were obtained from Shanghai Sino-British SIPPR/BK Lab Animal Co., Ltd. (China) and they were maintained in controlled condition as our previous study described (*Jian et al., 2018*). All animal experiments for the study followed the Guide for the Care and Use of Laboratory Animals was approved by the Animal Ethics Committee of China Pharmaceutical University (certificate number: SYXK2016-0011, approval date: 27 January 2016 to 26 January 2021).

## Experimental protocol
At first stage, mice were provided either a normal chow diet (10% of calories) or a high fat diet (60% of calories) for 4 weeks, which were obtained from Shanghai Lab-oratory Animal Co. Ltd. (Shanghai, China). Then, the high fat diet-fed mice were randomly divided into 3 groups: one control group and two test groups, the test groups were administered with low dose of TQPE (15 mg/kg) and high dose of TQPE (30 mg/kg) per day. Normal control groups (CON) and high fat diet-fed mice control (HFD) group were treated with solution of 0.5% CMC-Na. All mice in each group were administrated relevant reagents for another 8 weeks. In the end of experimental period, all the mice were fasted for 12 h, then anesthetized and euthanized by isoflurane gas. Thereafter, the serum was separated from blood and the liver was removed and weighted. All the samples were stored at −80 °C condition before use.

## Biochemical parameters analysis
According to the commercial kits instructions, serum TC, HDL, LDL, TG, SOD, MDA, ALT and AST concentrations were determined in a microplate reader (Molecular Device, CA,

USA). Serum insulin level was determined by using ELISA commercially kits. The HOMA-IR (Homeostatic model assessment for insulin resistance) value which was calculated as fasting blood glucose (mmol/L) × fasting serum insulin (mIU/L)/22.5, was represented insulin sensitivity.

### Liver pathological examination

Liver tissues were fixed overnight at 4 °C in 4% paraformaldehyde and embedded in paraffin. Sections (4 μm) were prepared for hematoxylin & eosin (H&E) stain. The histological images were observed and photographed under an optical microscope (Carl Zeiss, Germany).

### Western blot analysis of liver tissue

The total protein from the liver tissues was extracted using a commercial kit. Proteins from the liver supernatant were quantified by a BCA kit. 25 μg of total proteins was separated on 8–10% SDS-PAGE gel then transferred onto a PVDF membrane from Millipore (MA, USA). The membrane was blocked for 2 h in 5% nonfat milk powder solution at room temperature. Then the membranes were incubated with primary antibody at 4 °C overnight. At the second day, the membranes were incubated with secondary antibody at room temperature for 1–1.5 h. The Membrane-bound antibodies complexes were noticed by ECL detection (Santa Cruz Biotechnology, CA, USA). Tanon 5200 image system (Tanon, Shanghai, China) was used to examine the signal. $\beta$-actin was employed to demonstrate the standard proteins quantity. ImageJ software (NIH) was applied to quantify the density of western blotting bands

### Statistical analysis

All results were expressed as mean ± SE from 10 mice. All statistical analyses were done through the GraphPad Prism Software(Version 7.02, CA, USA). One-way ANOVA with a post-hoc test was used for the statistical significance evaluation. $P < 0.05$ was indicated statistical significance.

## RESULTS

### Analysis of chemical constituents and content of TQPE by HPLC-QTOF/MS and HPLC-DAD

According to Folin-Ciocalteu phenol test, the most abundant compound in the TQPE was polyphenolics (hydrolysable tannins), the content of which was accounting for 91.7 ± 2.1% (gallic acid equivalents). To furtherly understand the potential bioactive components in TQPE, a HPLC-QTOF/MS assay was carried out. The base peak chromatogram of TQPE was illustrated in Fig. 1A. Eight phenolic compounds ($R_t = 9.45$, 23.07, 24.15, 24.76, 30.72, 35.58, 36.52 and 47.65 min) were tentatively identified according to previous studies (*Hatano et al., 1990*; *Huang et al., 2016*). Their structures were shown in Fig. 1B. The predominated polyphenols in TPQE was gallic acid and its derivatives, which included hydrolysable gallotannins and gallates. We also used HPLC-DAD method to calculate the content of five polyphenols in TPQE, which was showed in Table 1. Gallic acid

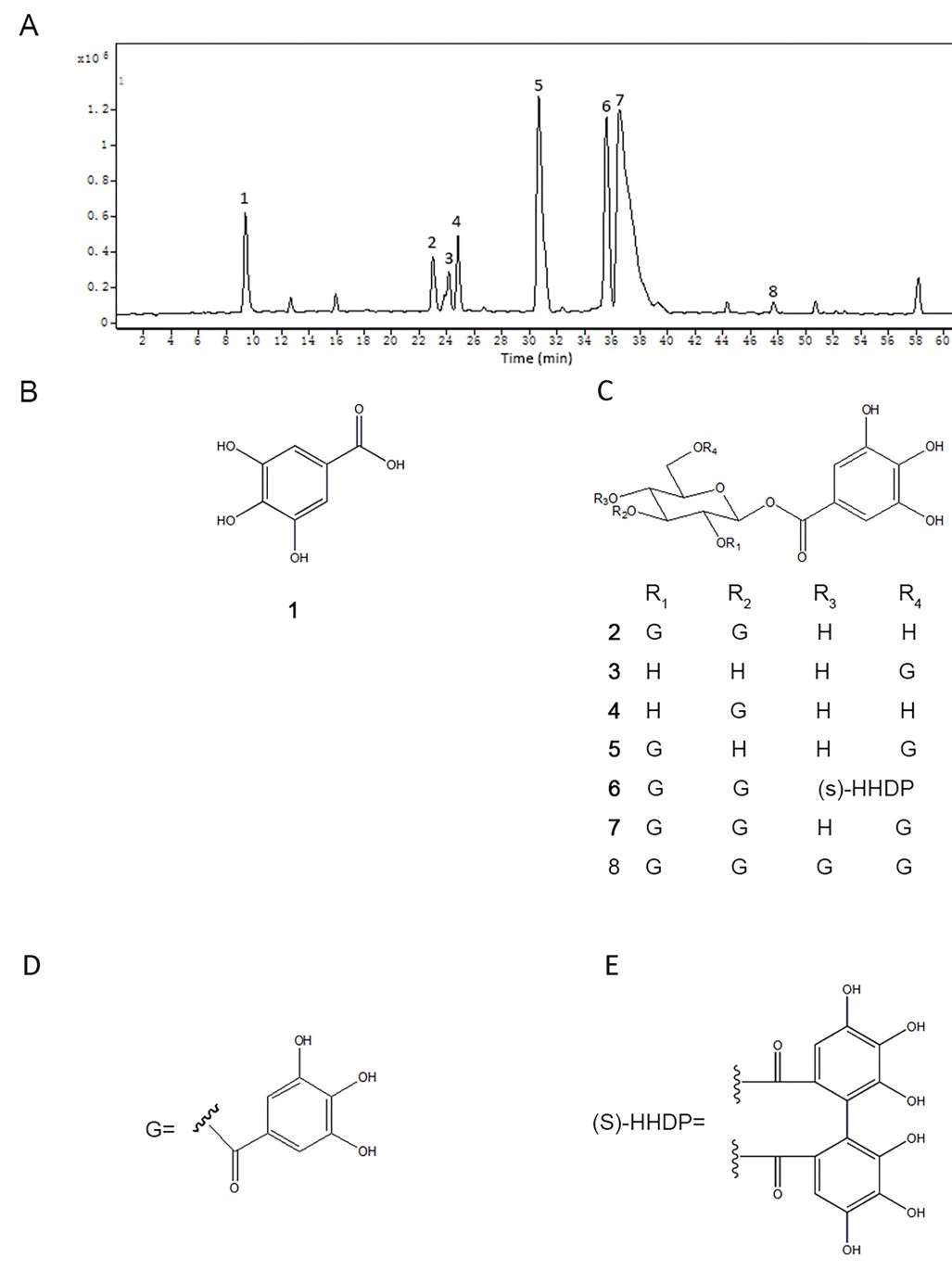

**Figure 1** **The chromatogram of TQPE analyzed by HPLC-QTOF (A) and the structures of eight dominated polyphenols in TQPE (B–E).**

(Compound 1), 1,2,3-tri-O-galloyl-β-D-glucopyranose (Compound 2), 1,2,3-tetra-O-galloyl-4,6-O-hexahydroxydiphenoyl-β-D-glucopyranoside (Compound 6), 1,2,3,6-tetra-O-galloyl-β-D-glucopyranose (Compound 7), 1,2,3,4,6-penta-O-hexahydroxydiphenoyl-β-D-glucopyranoside (Compound 8) were accounted for 696.56 ± 4.61mg/g.

**Table 1 Concentration of five polyphenols in TPQE by HPLC-DAD analysis ($n = 3$).**

| Peak number | Compound name | Concentration (mg/g) |
| --- | --- | --- |
| 1 | Gallic acid | $27.66 \pm 0.99$ |
| 2 | 1,2,3-tri-$O$-galloyl-$\beta$-D-glucopyranose | $19.89 \pm 0.07$ |
| 6 | 1,2,3-tetra-$O$-galloyl-4,6-$O$-hexahydroxydiphenoyl-$\beta$-D-glucopyranoside | $182.12 \pm 2.38$ |
| 7 | 1,2,3,6-tetra-$O$-galloyl-$\beta$-D-glucopyranose | $461.91 \pm 7.28$ |
| 8 | 1,2,3,4,6-penta-$O$-hexahydroxydiphenoyl-$\beta$-D-glucopyranoside | $4.04 \pm 0.09$ |

### Effect of TQPE on the body and liver weight of HFD mice

As showed in Fig. 2A and Fig. 2B, compared with the normal group, body weight in HFD group and their liver to body weight ratio rose obviously ($P < 0.001$). Administration of TQPE (15 and 30 mg/kg) significantly inhibited HFD induced growth in body weight and liver to body weight ratios ($P < 0.01$). Treatment with TQPE reversed the continuing weight gain in both body and liver.

### Histopathological examination

Normal control group livers showed glossy and resilient appearance. In contrast, as showed in Fig. 2C, the NAFLD mice livers were enlarged with yellow necrosis foci appearing. After the livers were treated with TQPE, the liver appearance improved in a dose-dependent manner (Fig. 2C).

Figure 2D displayed liver histology photo sections. In the control group, liver tissues showed normal liver lobular structure without fatty accumulation or inflammatory. In HFD treated mice, liver sections revealed significantly higher damage of hepatic lobule structures compared with normal mice, while lipid droplets were also spotted in liver cells. As displayed in Fig. 2D, after treating with TQPE in 15 and 30 mg/kg, hepatocyte swelling as well as quantities and volumes of lipid droplet were all alleviated. Morphology of liver lobular structure almost regained normal status, especially in the mice treated with 30 mg/kg TQPE.

### TQPE ameliorates lipid parameters in NAFLD mice

Relevant lipids parameters were detected to evaluate the effects of TQPE on lipid metabolism in NAFLD mice. TC, TG and LDL content were markedly elevated ($P < 0.001$; Figs. 3A, 3B and 3D) along with a decrease in HDL level (Fig. 3C; $P < 0.05$) in NAFLD mice compared to the normal group. In Figs. 3A–3C, the data showed that administration of TQPE (15 mg/kg) significantly reduced TC, TG and elevated HDL compared with the HFD group ($P < 0.01$). Furthermore, after treatment with a high dose of TQPE at 30 mg/kg, besides above effects, it dramatically decreased LDL but increased HDL level (Figs. 3A–3D; $P < 0.001$).

### TQPE changes the oxidative stress balance and reduces liver injury in NAFLD mice

As demonstrated in Fig. 4A and 4B, HFD group mice showed less adequate SOD and surplus level of MDA in the serum compared to the control group. This phenomenon

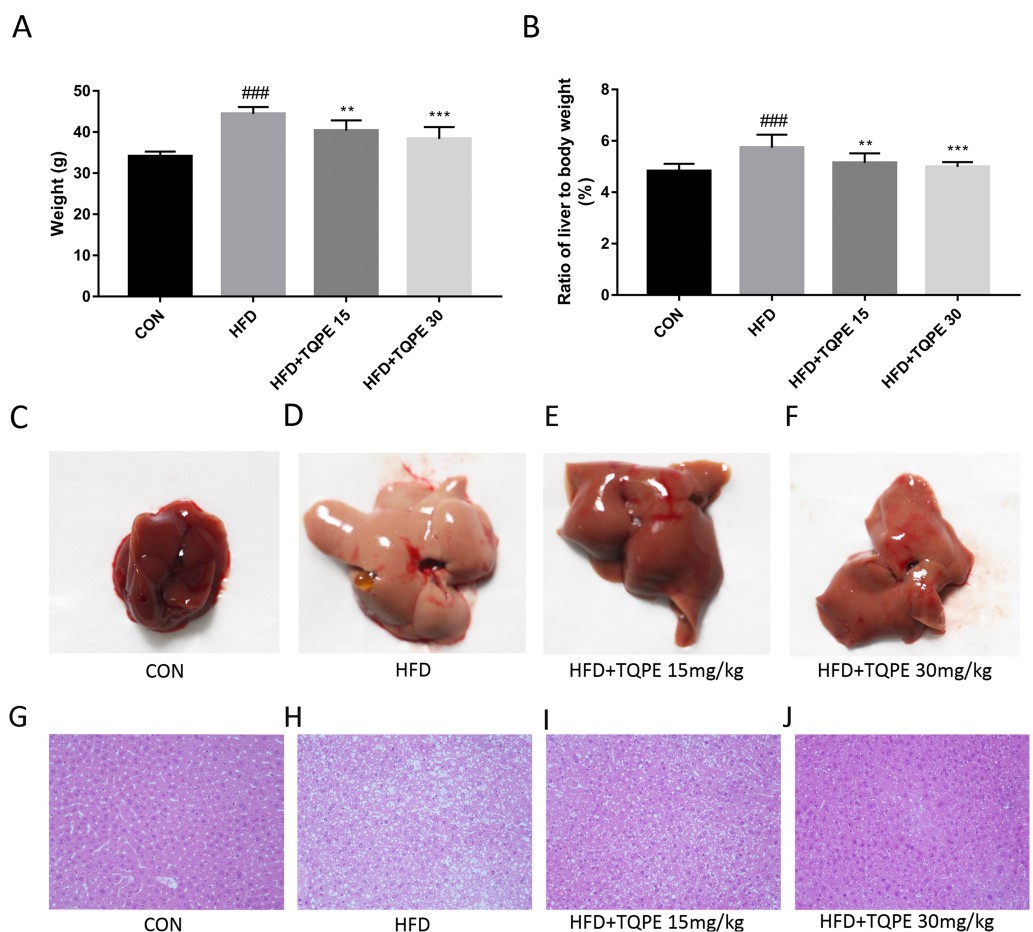

**Figure 2** Body weight (A), ratios of liver to body weight (%) (B), appearance of the liver (C–F), histological analysis of liver tissues (H & E, 200× magnification) (G–J). All the results were presented as mean ± SE ($n = 10$). ###$P < 0.001$ vs. the CON group; **$P < 0.01$, ***$P < 0.001$ vs. the HFD group.

implying an increased oxidative stress and a decrease of antioxidant capacity in HFD-induced NAFLD mice. Treatment with TQPE ameliorated this symptom by significantly raising SOD and reducing MDA level ($P < 0.001$, Figs. 4A and 4B). In Figs. 4C and 4D, compared with normal mice, the contents of ALT and AST were seen elevating in the NAFLD mice. Administration of TQPE (15 and 30 mg/kg) significantly reduced ALT and AST levels indicating the recovery from liver damage induced by HFD-induced NAFLD mice ($P < 0.001$).

## Effects of TQPE on serum insulin and Homeostatic Model Assessment of Insulin Resistance (HOMA-IR)

Insulin concentration in serum and HOMA-IR value were calculated to evaluate the effect of TQPE on insulin blood level and action.

In HFD mice, both the serum insulin concentration and HOMA-IR value were increased significantly, which indicating that there was an insulin resistance in NAFLD mice

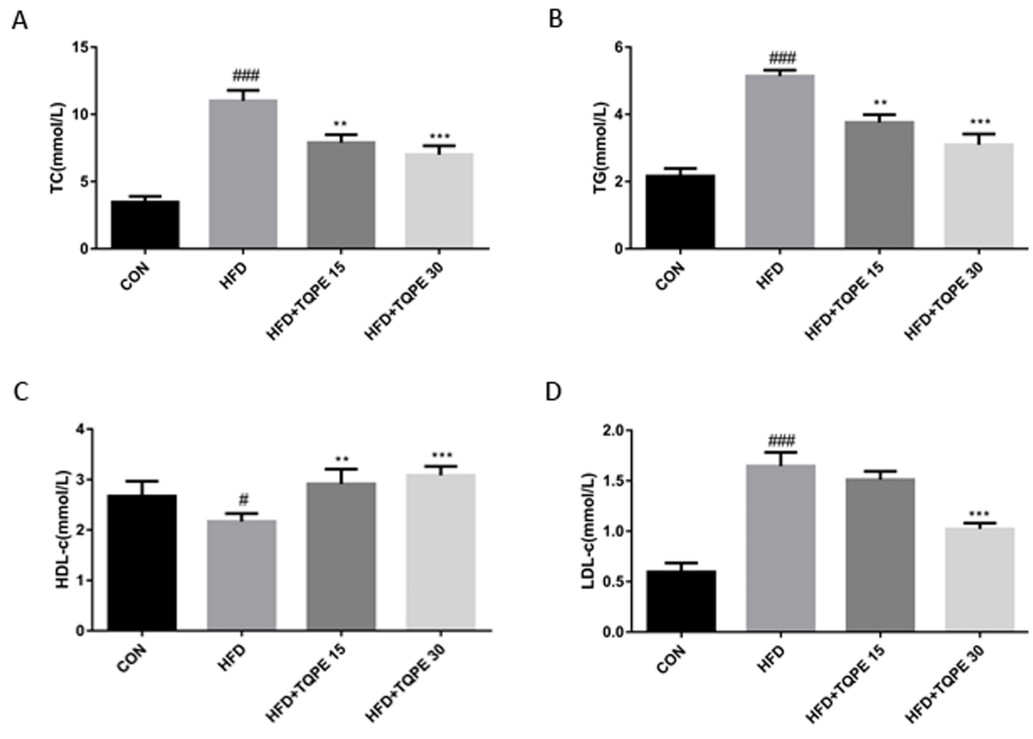

**Figure 3** **Effects of TQPE in HFD-induced NAFLD on plasma total cholesterol (TC) (A), triglycerides (TG) (B), high-density lipoprotein cholesterol (HDL) (C), and low-density lipoprotein cholesterol (LDL) (D) were measured.** All the results were presented as mean ± SE ($n = 10$). $^{\#}P < 0.05$, $^{\#\#\#}P < 0.001$ vs. the CON group; $^{**}P < 0.01$, $^{***}P < 0.001$ vs. the HFD group.

($P < 0.001$, Fig. 5). Both the value of serum insulin concentration and HOMA-IR were significantly decreased after the treatment of TQPE (15 and 30 mg/kg) in comparison with the HFD mice ($P < 0.001$, Figs. 5A and 5B). The data indicated that TQPE suppressed HFD-induced IR in NAFLD mice.

## TQPE changes pathways participated in lipid metabolism and insulin resistance

For further investigate the mechanism of TQPE preventing NAFLD, we detected the change of proteins expression including AMPK, SREBP, ACC, IRs-1 and Akt in the liver tissue, which were involved in lipid metabolism and insulin resistance. In Figs. 6A–6C, the phosphorylation of AMPK and ACC were decreased along with an increase in SREBP in HFD group in comparison with control group ($P < 0.001$). Meanwhile, compared with control group, in HFD mice liver, it showed a drop of IRs-1 and Akt phosphorylation ($P < 0.001$; Figs. 6D and 6E). Treatment with TQPE at a high dose of 30 mg/kg markedly recovered the phosphorylation level of AMPK, ACC and inhibited the expression of SREBP in HFD-induced NAFLD mice ($P < 0.001$; Figs. 6A–6C). Additionally, TQPE (15 and 30 mg/kg) administration kept the similar level of IRs-1 and Akt phosphorylation than the detected in the control mice ($P < 0.01$; Figs. 6D and 6E).

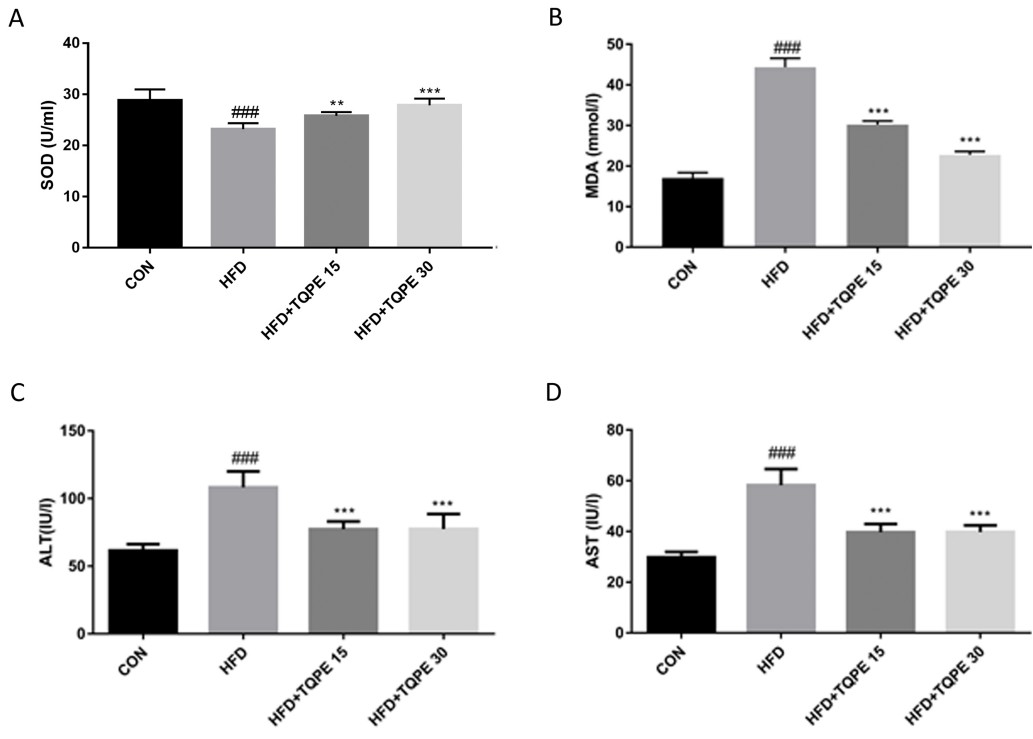

**Figure 4 TQPE administration inhibits oxidative stress and liver injury in NAFLD.** The levels of SOD (A), MDA (B), ALT (C) and AST (D) in the serum were measured. All the results were presented as mean $\pm$ SE ($n = 10$). $^{###}P < 0.001$ vs. the CON group; $^{***}P < 0.001$ vs. the HFD group.

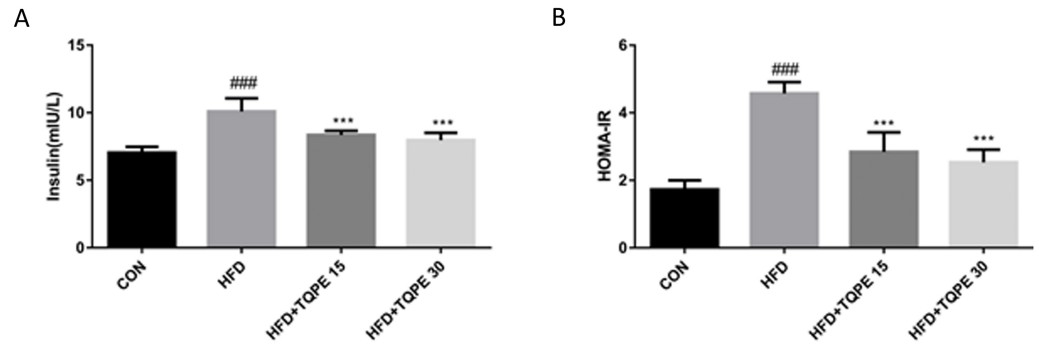

**Figure 5 Effects of TQPE on insulin and HOMA-IR in HFD-induced NAFLD.** The levels of insulin (A) in the serum and HOMA-IR value (B) were measured. All the results were presented as mean $\pm$ SE ($n = 10$). $^{###}P < 0.001$ vs. the CON group; $^{***}P < 0.001$ vs. the HFD group.

## DISCUSSION

The current study investigated the therapeutic effect of *Trapa quadrispinosa* pericarp extract (TQPE), on lipid accumulation and insulin resistance in HFD-induced NAFLD, and worked out the possible mechanism. We concluded that TQPE attenuated HFD-induced lipid accumulation in the liver, probably was mediated by the AMPK activation

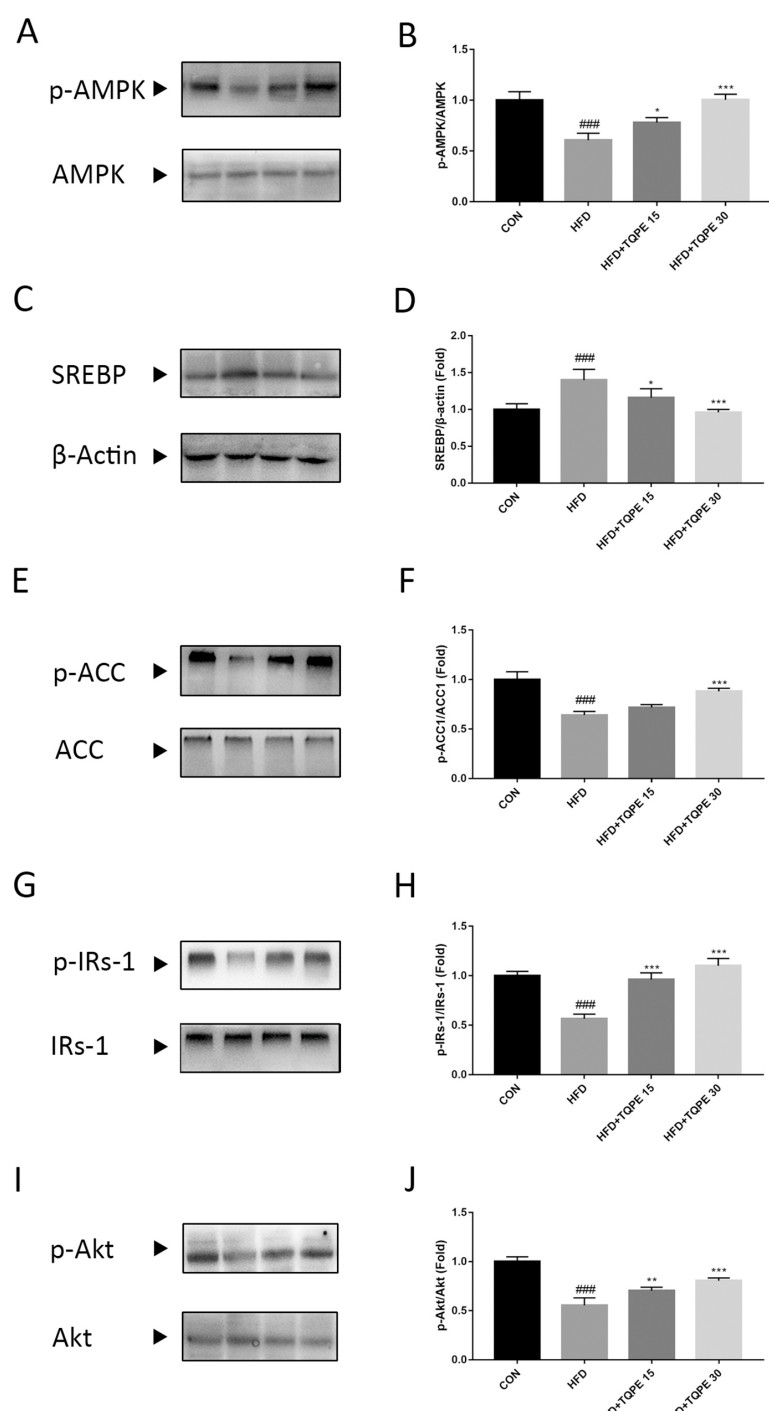

**Figure 6  Effects of TQPE on AMPK, SREBP, ACC, IRs-1 and Akt in the liver tissue of HFD-induced NAFLD mice.** p-AMPK, AMPK (A–B), SREBP (C–D), p-ACC, ACC (E–F), p-IRs-1, IRs-1 (G–H), p-Akt and Akt (I–J) expression were probed by western blotting. Results were presented as mean ± SE ($n = 3$).
[###]$P < 0.001$ vs. the CON group; [*]$P < 0.05$, [**]$P < 0.01$, [***]$P < 0.001$ vs. the HFD group.

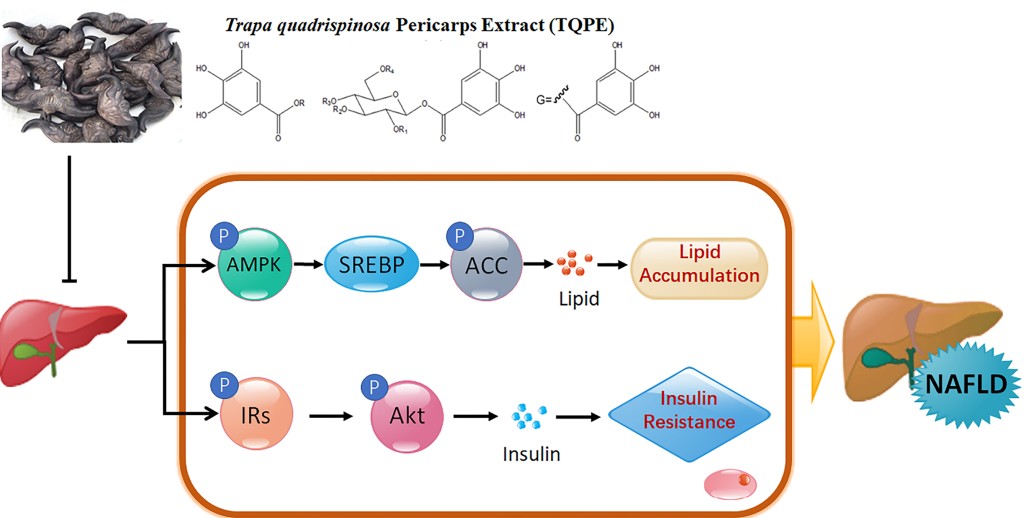

**Figure 7** A schematic illustration of how TQPE interfered with lipid accumulation and IR in HFD-induced NAFLD mouse.

and SREBP-mediated lipogenesis inhibition. TQPE also improved IR in NAFLD possibly through upregulating the levels of p-IRs-1 and p-Akt protein (Fig. 7).

It is widely accepted and confirmed that hepatic steatosis is the earliest stage of NAFLD, which is manifested as accumulation of lipid in the liver. In nutrient oversupply states, ectopic lipid accumulation is closely related to hyperglycemia and hyperlipidemia. AMPK is a crucial key and important metabolic sensor and regulator of lipid and glucose metabolism in diverse tissues and cells (*Cheng et al., 2016*; *Day, Ford & Steinberg, 2017*). Activated AMPK promotes energy production, meanwhile, and represses ATP-consuming processes, to maintain energy status balance (*Gonzales, Garcia-Hjarles & Velasquez, 1992*; *Ruderman & Prentki, 2004*). Both SREBP and ACC are two downstream effectors of AMPK. For instance, AMPK activation inhibits fatty acid synthesis via a reduction in the transcriptional activation of SREBP, which is a key transcription factor during *de novo* lipogenesis in the regulation of lipogenic genes including *acc1* in the liver (*Ahmed & Byrne, 2007*; *Zhou et al., 2017*). AMPK activation also could switch off fatty acid synthesis through ACC phosphorylation in adipose and hepatic tissues (*Bijland, Mancini & Salt, 2013*; *Zhang, Xie & Leung, 2018*). As described, AMPK activation might reduce hepatic lipid accumulation via lipogenesis inhibition, indicating that AMPK is a potential therapeutic target for the treatment of hepatic steatosis in NAFLD (*Zhou et al., 2017*). Recent studies showed that many compounds are able to activate AMPK in animal and cellular models to improve lipid metabolism in NAFLD, by inhibiting SREBP and increasing phosphorylates of ACC (*Chen et al., 2019*; *Kang et al., 2019*; *Park et al., 2019*). In our study, we explored the lipid contents comprising TC, TG and LDL, the concentration of which were significantly lowered after TQPE treatment, while the HDL level was enhanced in HFD-induced NAFLD ($P < 0.01$; Fig. 3). We also found that TQPE at dose of 30 mg/kg/day could obviously activate the phosphorylation of AMPK and ACC, while inhibit SREBP expression ($P < 0.001$;

Figs. 7A–7C). These results demonstrated that AMPK axis is associated with the protective effects of TQPE in lipid metabolism under NAFLD status.

Impaired responsiveness to insulin evokes insulin resistance in NAFLD, hence, the improvement of insulin sensitivity is important for the treatment of NAFLD. Insulin receptor substrate-1 (IRs-1) is a core factor in insulin signaling transduction. Research showed that phosphorylation of IRs-1 at Ser accounts for IR, while phosphorylation of IRs-1 at Tyr is required for responses of insulin stimulates (*Bhattacharyya, Feferman & Tobacman, 2015*; *Xiao et al., 2018*). In metabolic disorders, such as NAFLD and T2DM, impaired IRs-1 level was observed, for instance, lipid accumulation in the liver was closely associated with increased serine phosphorylation of IRs-1 (*Araujo et al., 2018*; *Dallak, 2018*; *Dong et al., 2019*). Restoration of IRs-1 might be a beneficial treatment for curing NAFLD (*Yang et al., 2018*; *Zhou et al., 2018*). Protein kinase B (Akt) as downstream of IRs-1, plays an essential role in insulin signaling cascade. In response to upstream signal of IRs-1 on Tyr, activated Akt increases hepatic glucose uptake, glycogen synthesis and decreases lipogenesis (*Cignarelli et al., 2019*; *Geidl-Flueck & Gerber, 2017*; *Guo, 2014*). Moreover, hepatic Akt activation enhances the insulin sensitivity (*Cignarelli et al., 2019*; *Ke et al., 2015*). Therefore, IRs-1/Akt is important for hepatic insulin signaling, regulates this pathway might be beneficial for treating NAFLD. In the current work, the level of insulin and HOMA-IR value increased significantly in HFD-induced NAFLD ($P < 0.001$; Fig. 5). This mechanism supported by decreased hepatic phosphorylation of IRs-1 on Tyr ($P < 0.001$; Fig. 6D). In parallel, there was also a significant decline in phosphorylation of Akt ($P < 0.001$; Fig. 6E) in NAFLD mouse model. TQPE administration with different doses (15 and 30 mg/kg/day) reversed the high level of insulin and HOMA-IR value compared with NAFLD mice ($P < 0.001$; Fig. 5). This effect might be associated with restore phosphorylation of IRs-1 and Akt ($P < 0.05$; Figs. 6D and 6E) by TQPE to improve IR in NAFLD model.

In addition, ALT and AST are two hepatic enzymes, which are well-known key biochemical markers for detecting liver damage in NAFLD mode (*Katsagoni et al., 2017*). In our study, HFD treatment induced increasing of ALT and AST, while the administration of TQPE (15 and 30 mg/kg/day) could revise these parameters ($P < 0.001$; Figs. 4C and 4D). In comparison with a dose of 2,000 mg/kg of *T. natans* pericarp extract displaying antihyperglycemic effect in rats (*Yasuda et al., 2014*), and based on exploratory studies of 50–100 mg/kg doses, we found that a lower dose of 30 mg/kg of TQPE, also exerted a therapeutic action on HFD-induced NAFLD in mice, while no significant abnormal changes were found in the present study, so we stepped down doses for this study to provide a good safety margin. According to USA Food and Drug Administration's instruction (*Food and Drug Administration, 2005*), the dose of 30 mg/kg TQPE used in human is approximately 200 mg/day, thus this dose is acceptable for patients compared to other clinical trials used (250–960 mg/day) (*Chen et al., 2016*; *Qiu et al., 2013*; *Ried, Frank & Stocks, 2013*).

## CONCLUSIONS

The pericarps of *T. quadrispinosa* is a kind of agricultural waste, that are usually discarded in large quantities after the seeds had been harvested. From economic aspect, the present

study helps to discover polyphenol-rich extraction from this waste with medicinal value. The present study demonstrated that *Trapa quadrispinosa* pericarp extract (TQPE) administration significantly improve the metabolic parameters including the insulin resistance in high-fat diet-induced NAFLD mice. The protective mechanisms of TQPE may be attributed to regulate AMPK/SREBP/ACC and IRs-1/Akt signal pathways, which are involved in lipid metabolism and IR, respectively. TQPE treatment provides a novel therapeutic strategy to prevent HFD-induced NAFLD, however, there were still some limitations in this study (i) the amount of food eaten was not directly measured, however, observation by caretakers showed that the animals were feeding and the loss of weight was not simply related to decreased eating; (ii) the active entity in the extract which is absorbed and found in the plasma of the mice was not determined. In further studies, we warranted to determine the bioavailability of the active entity, and evaluate the mechanisms by which polyphenolic mediates AMPK/SREBP/ACC and IRs-1/Akt in HFD-induced NAFLD.

## ACKNOWLEDGEMENTS

The authors would like to thank Mr. Shirong Yang (Centre for instrumental analysis, College of Forestry, Nanjing Forestry University, Nanjing, China) for his useful help in the compound structure analysis.

### Funding

This research was funded by the National Natural Science Foundation of China (No. 81703224; No. 81773885; No. 31770366), The Jiangsu Province Science and Technology Modern Agricultural Plan (No. BE2016383), the Jiangsu Province Independent Innovation in Agricultural science and technology fund (CX(18)3042), and the Natural Science Foundation of the Jiangsu Province (BK20160603). Support was also received from Jiangsu Scientific and Technological Innovations Platform (Jiangsu Provincial Service Center for Antidiabetic Drug Screening) and the Jiangsu Key Laboratory for the Research and Utilization of Plant Resources (No. JSPKLB201825; No. JSPKLB201833). The funders had no role in study design, data collection and analysis, decision to publish, or preparation of the manuscript.

### Grant Disclosures

The following grant information was disclosed by the authors:
National Natural Science Foundation of China: 81703224, 81773885, 31770366.
Jiangsu Province Science and Technology Modern Agricultural Plan: BE2016383.
Jiangsu Province Independent Innovation in Agricultural science and technology fund: CX(18)3042.
Natural Science Foundation of the Jiangsu Province: BK20160603.
Jiangsu Scientific and Technological Innovations Platform (Jiangsu Provincial Service Center for Antidiabetic Drug Screening).

Jiangsu Key Laboratory for the Research and Utilization of Plant Resources: JSPKLB201825, JSPKLB201833.

## Competing Interests

The authors declare there are no competing interests.

## Author Contributions

- Tunyu Jian performed the experiments, prepared figures and/or tables, authored or reviewed drafts of the paper, approved the final draft.
- Han Lü conceived and designed the experiments, performed the experiments, prepared figures and/or tables, authored or reviewed drafts of the paper, approved the final draft.
- Xiaoqin Ding performed the experiments, authored or reviewed drafts of the paper, approved the final draft.
- Yuexian Wu performed the experiments, prepared figures and/or tables, approved the final draft.
- Yuanyuan Zuo and Jiawei Li analyzed the data, prepared figures and/or tables, approved the final draft.
- Jian Chen conceived and designed the experiments, authored or reviewed drafts of the paper, approved the final draft.
- Hong Gu analyzed the data, contributed reagents/materials/analysis tools, prepared figures and/or tables, approved the final draft.

## Animal Ethics

The following information was supplied relating to ethical approvals (i.e., approving body and any reference numbers):

All animal experiments for the study followed the Guide for the Care and Use of Laboratory Animals was approved by the Animal Ethics Committee of China Pharmaceutical University (certificate number: SYXK2016-0011).

## Data Availability

The raw measurements are available in the Supplemental Files.

## Supplemental Information

Supplemental information for this article can be found online at http://dx.doi.org/10.7717/peerj.8165#supplemental-information.

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
