# Peer review of "Polyphenol-rich Trapa quadrispinosa pericarp extract ameliorates high-fat diet induced non-alcoholic fatty liver disease by regulating lipid metabolism and insulin resistance in mice"

_PeerJ, doi:10.7717/peerj.8165_

## Round 0.1 · original submission · Major Revisions

· Academic Editor

Major Revisions

A discussed with the Editorial Office, some essential clarification was needed. You provided what seems to be a reasonable complement of information. However, this must now be introduced in your paper. Please, provide a revised version in which you introduce the key points of your answer in the Discussion section (or elsewhere if more appropriate) and critically discuss whether a dose of 30mg/kg in rat would, after allometric scaling, correspond to a human dose (state this dose) that would be medically (toxicity issues ?), practically (can patients swallow that ?) and economically (would this dose no be so costly that no one is going to take it ?) acceptable. I *do* realize that the last point (economic aspect) may be very difficult to discuss, since the cost of final medication could be considerably lower (or higher) than what is being used in your study but some discussion about this point would nevertheless be useful as it is a key element in a "go/no-go" decision of most chemotherapeutic programs.

Once you have uploaded your revised version, I'll look at it and and initiate the reviewing process if the additions/corrections have been made as suggested.

I kook forward to information/instructions from you about this revised (post-editorial initial assessment) version.

---

## Round 0.2 · Major Revisions

· Academic Editor

Major Revisions

As you can see, your paper was considered as interesting and valuable, but a number of comments and critiques need to be addressed. I'd like my-self to insist on the necessity to discuss the dosages used and to convince the reader that these dosages are meaningful if considering potential human therapy. One of the reviewers suggest hints in this connection but you may have other suggestions. In any case, we *must* be able to distinguish between what could be nice but somewhat gratuitous biological studies and studies that clearly show that the compounds discussed are amenable to use in humans. This includes dosages and potential toxicities.

Please, submit a detailed rebuttal where I can see clearly where and how you have taken all comments and suggestions into consideration. If you do not agree with some of these comments or suggestions, explain why. Your rebuttal will be an essential element for me to make a final decision on your paper. Please, note also that your revised version may enter a new round of review by the same or by different reviewers. I cannot, therefore, make any commitment about a final acceptance of your submission. .

·

Basic reporting

There are no concerns on basic reporting and language

Experimental design

As noted in the review, some points such as pharmacokinetics to determine the active component and dose justification is not reported

Validity of the findings

The data partially supports the findings but the deficiency is that w e do not know which component is active.
Also if the animals simply stopped eating the high fat food.

Additional comments

Reviewer Comments.

There is an urgent need for adequate treatment for the growing NAFLD and hepatocellular carcinoma. This manuscript reveal a natural product extract that could be beneficial.
Simple editing:
All “et al. “ in the manuscript should be in italics.
Please search the entire manuscript and change.
Line 52. Replace restore with treat and prevent
Line 72. in the south of China, as well as in India and south east Asia.
Line 96. Five standard (lower case s)
Line 107: Delete The primary antibodies and replace simply with Antibodies to
Line 115: Change concentrated to concentrating
Line 112: Define TQPE as Trapa quadrispinosa pericarp extract. Also, the name of the plant must always be in italics.
Line 196. Please change dominated to predominant
Line 239. Define HOMA-IR. Homeostatic Model Assessment of Insulin Resistance.
Line 321. Please change to used in other clinical trials.
Line 332. ..waste, that are usually discarded
Questions that need to be answered:
Line 153. How do we know that the mice continue to eat the high fat diet after beginning treatment? It is possible that they lost their appetite and ate less. Was the amount of food eaten (or left in food tray) measured?
Line 153. How ere these doses selected. Normally one selects does based on blood levels and scaled down from toxic limits
In the introduction the authors should note if the pericarp of TQ is used in China as natural medicine for liver disease. Is this used what led the authors to identify the active components?
Also, there are other Trapa species (such as T. japonica and others. Is the activity specific to TQ?
Was there any plasma pharmacokinetic analysis on the TQPE conducted? Steady state plasma levels are essential. It will also show which of the components are actually absorbed. Of the several constituents of the TQPE it is possible that in the intestinal tract some of them may be converted and it is also likely that only one or more components are actually absorbed. What is the oral bioavailability of each component? This is important and needs to be determined prior to publishing this manuscript. The authors may be surprised that only one or more components are needed and also that there may be conversion in the gut.
Line 245. Was there a dose response between the 15 and 30 mg/kg doses tested in each of the parameters?
It would be helpful to have a Figure of the proposed NAFLD signaling pathways. I note the pathway as the last Figure. That Figure also needs clarification. There are several up and down arrows. Which of these are related to the TQPE treatment. It is not clear to me.

·

Basic reporting

The authors showed reduction of the diet induced NAFLD associated to obesity in ICR mice due to administration of a specific herb extract (TQPE), from the pericarp of an aquatic plant popular in south of China, which fruit is used as folk medicine and functional food. They also suggested that AMPK/SREBP/ACC intracellular pathway in the liver is involved in the detected effects.

The authors performed:
1- HPLC from the TQPE
2- liver histology by HE from the DIO with NAFLD mice;
3- body and liver mass;
4- AST, ALT, TG, TC, HDLc, LDLc, SOD, and MDA in blood samples;
4- WB in liver samples of AMPK, SREBP, and ACC.

The language of the manuscript is clear. The references provide sufficient background to its proposal.
The figures and tables are well done and convincing.
The raw data is shared. However, it should be interesting for readers to know which animal group corresponds to each band in the supplied PVDF membranes.
The hypothesis is clear and the data presented is adequate to confirm it.

Experimental design

The subject of the manuscript is within the scope of the journal.
The hypothesis is clearly defined and the aims of the study are adequately indicated within the research field.
The ethical approval for animal usage is described in the manuscript, however, I think it should include the following information in the Methods section and or in the legend figures and tables:
1- the total number of animals used in each parameter analyzed;
2- the entire period of HFD intake;
3- the via of administration of the TQPE;
4- if the animals were euthanized after any period of fast or in fed condition.

Validity of the findings

The data presented is convincing. The statistical analysis performed is adequate, however there are 3 minor points that I would like to hear from the authors:
1- Is there any effect of the TQPE in control chow animals, such as reduced insulinemia or body weight?
2- Is there any impact of the TQPE treatment on food and or water intake?
3- Are the elevated AST and ALT blood levels associated to any other hepatic injury histological marker?

Additional comments

The manuscript is very easy to read and the figures are well presented.
The inclusion of the translational information regarding the equivalent dose for human is positive, as well as the economical aspect of identifying the potencial beneficial usage of the pericarp since it is an agriculture wast.
However, I would like to see included in the manuscript some information regarding the above mentioned 7 points.

---

## Round 0.3 · Minor Revisions

· Academic Editor

Minor Revisions

Your revised version was well received by the reviewers. However, some corrections are still needed. If you are ready to make the necessary changes (see the reports and the annotated PDF file submitted by reviewer 1), I'll be happy to see your submission again. Please, include a rebuttal explaining how (and where) you have taken the reviewers' comments into consideration.

·

Basic reporting

The Authors have provided reasonable responses to questions and have made the edits requested.
However, it is also required that some of the clarifications are added into the text for the readers, such as the determination of bioavailability of the active entity in future studies.

Experimental design

The experimental design is fine except that the reason for choosing the doses need to be added

Validity of the findings

The results presented are valid except that the authors must add that the mice simply decreased the amount of high fat diet they ate because of the treatment they were receiving. The food intake and the treatment dose intake was not measured so some estimate of what was observed that the animals were eating normally should be mentioned.

Additional comments

Same as above
A couple more edits are marked on the attached manuscript. It will be helpful if marked (Tracked edits) are returned so I do not have to red the whole paper again!

·

Basic reporting

The revised version of this manuscript is improved. However, there are some minor points I would like to be considered:

1- line 155: instead of "... sacrificed..." I suggest ... euthanized...

2- Please include how long the mice were fasted before euthanasia.

3- line 241: instead of "... insulin signal." I suggest to include ... insulin blood level and action.

4- line 257: instead of "... repaired prohibition of..."I suggest ... kept the similar level of IRS-1 and AKT phosphorylation than the detected in the control mice.

Experimental design

Please, include the number of hours the mice were fasted before euthanasia.

Validity of the findings

No comment.

---

## Round 0.4 · accepted · Accept

· Academic Editor

Accept

Thank you for making these final improvements. I believe this has made your paper stronger. I wish you good success with this publication and your future work.